# Effects of eicosapentaneoic acid on innate immune responses in an Atlantic salmon kidney cell line in vitro

Tor Gjøen[1]*, Bente Ruyter[2], Tone Kari Østbye[2]

**1** Department of Pharmacy, Section for Pharmacology and Pharmaceutical Biosciences, University of Oslo, Oslo, Norway, **2** Nofima (Norwegian Institute of Food, Fisheries and Aquaculture Research), Ås, Norway

* tor.gjoen@farmasi.uio.no

**Data Availability Statement:** SRA archive (https://www.ncbi.nlm.nih.gov/sra) with accession # PRJNA1033415.

**Funding:** TG, BR and TKØ received grant #901484 from the Norwegian Seafood Research fund (FHF)

## Abstract

Studies of the interplay between metabolism and immunity, known as immunometabolism, is steadily transforming immunological research into new understandings of how environmental cues like diet are affecting innate and adaptive immune responses. The aim of this study was to explore antiviral transcriptomic responses under various levels of polyunsaturated fatty acid. Atlantic salmon kidney cells (ASK cell line) were incubated for one week in different levels of the unsaturated n-3 eicosapentaneoic acid (EPA) resulting in cellular levels ranging from 2–20% of total fatty acid. These cells were then stimulated with the viral mimic and interferon inducer poly I:C (30 ug/ml) for 24 hours before total RNA was isolated and sequenced for transcriptomic analyses. Up to 200 uM EPA had no detrimental effects on cell viability and induced very few transcriptional changes in these cells. However, in combination with poly I:C, our results shows that the level of EPA in the cellular membranes exert profound dose dependent effects of the transcriptional profiles induced by this treatment. Metabolic pathways like autophagy, apelin and VEGF signaling were attenuated by EPA whereas transcripts related to fatty acid metabolism, ferroptosis and the PPAR signaling pathways were upregulated. These results suggests that innate antiviral responses are heavily influenced by the fatty acid profile of salmonid cells and constitute another example of the strong linkage between general metabolic pathways and inflammatory responses.

## Introduction

Immunometabolism is a rapidly expanding branch of immunology focusing on the effects of nutrients and energy metabolism on immune system function in health and disease [1]. The crosstalk between metabolic reprogramming and signal transduction in immune cells [2] and nonimmune cells during innate responses [3] are now under active investigations, both in vitro [4] and in vivo [5]. One striking example is the effect of glucose and lactate levels on poly I:C induced RLR signaling leading to changes in interferon secretion and viral replication [6]. This study showed that lactate interact directly and inhibit the mitochondrial antiviral signaling protein (MAVS) binding to the mitochondrial membrane, and therefore signaling.

https://www.fhf.no/ The sponsor played no role in the experimental design.

**Competing interests:** The authors have declared that no competing interests exist.

Another antiviral signaling protein (stimulator of interferon genes = STING) has recently been shown to interact with the fatty acid desaturase, Fads2, an enzyme important for PUFA synthesis. Upon STING activation (by microbial nucleotides), Fads2 is released and free to synthesize PUFA's important for STING inhibition and release of inflammation [7]. These are only two examples of the many reports on immunometabolism in mammalian systems. However, apart from zebrafish or aquaculture feeding studies combined with microbial challenge [5, 8–12] there are few reports on the mechanistic of the interplay between feed and immunity in fish [13, 14]. Fish oil and fish meal have traditionally been the major dietary source of long chain n-3 polyunsaturated fatty acids eicosapentaneoic acid (EPA) and docosahexaenoic acid (DHA) in diets of Atlantic salmon (*Salmo salar)*. Increasing demands for EPA and DHA as ingredients in human health products, pet food and fish feed have resulted in shortages of these fatty acids on the international markets. Increasing substitution of fish oil by plant oils in Atlantic salmon feed has been one response, thereby reducing the EPA and DHA content in salmon [15].

EPA and DHA are both believed to be essential nutrients in salmon diets, they are needed to secure good growth, health, and quality [16]. The dietary requirement for EPA and DHA for maintaining good growth is estimated to 6% of total fatty acids in the diet [17]. However, when the salmon experiences demanding environmental conditions in sea cages, the requirement for these essential fatty acids is even higher [18]. EPA and DHA have different physiological functions in the body. Among the many functions, DHA for instance are a major constituent of membrane phospholipids and EPA are known to be an important precursor to eicosanoids. Whether salmon has a specific dietary requirement for EPA alone is still unknown, since all requirement studies are based on using fish oil which contains both EPA and DHA. Varying the dietary levels of EPA and DHA has been shown to alter fatty acid composition of phospholipids in tissues of several fish species [19].

It is well known that EPA and DHA can modulate inflammation and immune responses both in humans and experimental animals, mainly with anti-inflammatory effects [20, 21]. These anti-inflammatory effects can modulate the severity and outcome of infections and autoimmune diseases [22], and some studies imply that EPA and DHA can improve host resistance to pathogens [23, 24]. Human PBMCs incubated with EPA and DHA in vitro display a clear dose related inhibition of proinflammatory cytokine expression (IL-2 and TNF-a) in T-helper cells, with EPA as the most potent inhibitor [25]. Investigations regarding this topic have yet to provide a clear conclusion [26]. In vitro experiments have demonstrated that replacement of fish oil by vegetable oils impairs macrophage function (e.g. phagocytic and respiratory burst activity) in Rainbow trout [27], Gilthead sea bream [28, 29], European sea bass [30], and Channel catfish [31]. Others have reported no effects on immune cell function with this replacement in Atlantic salmon [32, 33]. Reports also suggests that high dietary EPA content can reduce heart inflammation and pathology in Atlantic salmon following viral challenge [13, 34, 35].

Poly I:C is a double stranded, synthetic RNA molecule eliciting a transient antiviral state in vitro and in vivo in vertebrates [36] and invertebrates [37] mainly through potent interferon induction. Injection of poly I:C in fish confers resistance to viral infections lasting up to several weeks [38–40]. After injection or stimulation of fish cells in culture, hundreds of interferon stimulated genes (ISGs) are upregulated, supporting the antiviral state [41–43]. Poly I:C has therefore been investigated a potential therapeutic [44, 45] and as a vaccine adjuvant in many experimental systems [46–48]. Poly I:C acts through a set of pathogen recognition receptors (PRRs), either transmembrane (TLRs) or cytosolic (RIG-I) proteins, resulting in altered transcription [49, 50]. It has previously been shown that EPA and DHA can modulate TLR function through interfering with lipid rafts and signaling platforms in cell membranes of immune

cells [21, 51, 52]. In addition, several reports link TLR activation to eicosanoid production through phospholipase A2 (PLA2) activation and increased cyclooxygenase-2 (COX-2) transcription [53]. Short and long term effects of n-3 PUFAs on innate immunity in immune and non-immune cells of mammals have been extensively studied [54]. In general, the anti-inflammatory effects of these FAs observed in vitro can be recreated in vivo in mouse models [55, 56]. However, dietary studies of FA effects on immunity in humans have given mixed results [57]. Caution is therefore required when extrapolating from cell cultures to live animals [54].

The main objective of this study was to test the hypothesis that various cellular EPA levels modulate the antiviral transcriptional responses in salmon cells during stimulation with poly I: C. Our results suggests that cellular levels of EPA markedly skews the transcriptional responses induced by a proinflammatory stimuli like poly I:C and therefore may be of importance when establishing dietary requirements for this fatty acid.

## Materials and methods

### Cell culture

The Atlantic salmon kidney cell line (ASK) [58] were kindly provided by Knut Falk (Norwegian Veterinary Institute). Cells were routinely split once a week (1:2) and routinely cultivated at 20°C in Leibovitz L-15 medium supplemented with 4 mM L-glutamine (both reagents from Lonza Biowhittaker, Verviers, Belgium), 10% fetal bovine serum, 40 µM 2-mercaptoethanol (both from Gibco, Life Technologies, Bleiswijk, The Netherlands) and gentamicin (50 µg/ml—Lonza Biowhittaker, Walkersville, USA). Cells were placed at 15°C one week before the experiment started, for acclimatization, and for all the duration of the experimental period. No experimental animals were used in this study.

### Experimental design

ASK cells were incubated at 15°C for a week before the experiment started. Cells were seeded in 14 35 mm wells (6-well plates) at a density of 1.5 x $10^5$ cells per well, (passages 40–50) and left overnight to adhere at 15°C. At study start, EPA in complex with bovine serum albumin (BSA) was added to duplicate wells at increasing concentrations from 0 to 200 uM EPA/ml (0-25-50-100-200-200 uM). After one week, wells containing 0-25-50-100-200 uM EPA were supplemented with 30 ug/ml poly (I:C) to stimulate innate immune responses. Some wells without added EPA and an extra set of wells with the highest (200 uM) EPA concentrations were used as unstimulated controls. After 24 h with poly (I:C) stimulation, wells were washed three times with PBS, lysed with RT buffer (QIAGEN, Hilden, Germany) and samples stored at—20°C until RNA isolation and analysis. The experiment was repeated 3 times resulting in six technical replicates (wells) for the analysis (n = 6).

### Fatty acid analysis of ASK cells

Total lipids were extracted from Atlantic salmon head kidney cells (grown in 75 cm2 flasks), using the method described by Folch et al. 1957 [59] and described in [60]. The chloroform phase was dried under nitrogen gas and the residual lipid extract was re-dissolved in benzene and trans-methylated over night with 2,2-dimethoxypropane and methanolic HCl at room temperature, as described by Mason and Waller 1964 and by Hoshi et al. 1973 [61, 62]. The methyl esters of fatty acids were separated in a gas chromatograph (Hewlett Packard 6890) with a split injector, using a SGE BPX70 capillary column (length 60 m, internal diameter 0.25 mm and thickness of the film 0.25 um) and flame ionization detector and HP Chem Station software. Helium was used as a carrier gas, and the injector and detector temperatures were

both 300˚C. The oven temperature was raised from 50 to 170˚C at a rate of 4˚C min-1 and then raised to 200˚C at a rate of 0.5˚C min-1. The relative quantity of each fatty acid present was determined by measuring the area under the peak corresponding to that fatty acid.

## Total RNA isolation and sequencing

Total RNA was extracted using Rneasy Mini Kit (QIAGEN, Hilden, Germany) according to the manufacturer's tissue protocol. A step for removal of genomic DNA was included by adding Dnase I (Rnase-Free Dnase Set, QIAGEN, Hilden, Germany). RNA was eluted in 50 µl Rnase-free distilled water. RNA concentration was measured using PicoDrop Pico100 (Pico-Drop Technologies, Cambridge, UK). For sequencing, 42 samples in total, were sent to Norwegian Sequencing Centre (NSC), where RNA quality were verified with Agilent 210e0 Bioanalyser (Agilent, USA), and library preparation was performed using TruSeqTM Stranded mRNA Library Prep Kit (Illumina Inc., San Diego, USA). Libraries were sequenced on Illumina HiSeq 4000 sequencer resulting in 150-bp paired-end reads.

## Quantitative PCR (qPCR)

RNA was reverse transcribed to cDNA using high-capacity RNA-to-cDNA kit (Applied Biosystems Inc., United States), following manufacturers protocol. qPCR was performed in 96-well plates on LightCycler 480 using SYBR Green Master Mix (both from Roche Diagnostics, Basel, Switzerland). Cycling conditions were: 95˚C for 5 minutes, 40 cycles of 95˚C for 10 s, 60˚C for 10 s and 72˚C for 10 s. Melting curve was measured by 95˚C for 5 s and 65˚C for 1 min. All qPCR experiments were performed using three experimental and two technical replicates. Cycle threshold (Ct) values were obtained and used to calculate correlation with RNA-seq. For calculation of relative expression levels, delta-delta Ct method was used [63]. 18s and ef1a were used as reference genes [64]. Primers used are listed in Table 1.

## Bioinformatics and statistics

Fastq files containing reads from the RNA-seq were mapped to the Atlantic salmon genome (GCF_000233375.1_ICSASG_v2_genomic.fna) using the HISAT2-Stringtie pipeline [69, 70]

**Table 1. List of primers used in qPCR analysis.**

| Genes | Direction | Sequence 5′→3′ | Accession Number | Amplicon | Reference |
|---|---|---|---|---|---|
| ef1a | F | CACCACCGGCCATCTGATCTACAA | AF321836 | 77 | [64] |
| | R | TCAGCAGCCTCCTTCTCGAACTTC | | | |
| 18S | F | TGTGCCGCTAGAGGTGAAATT | AJ427629.1 | 61 | [65] |
| | R | GCAAATGCTTTCGCTTTCG | | | |
| ifna1 | F | CCTGCCATGAAACCTGAGAAGA | AY216594 | 107 | [66] |
| | R | TTTCCTGATGAGCTCCCATGC | | | |
| isg15 | F | ATGGTGCTGATTACGGAGCC | AY926456 | 151 | [67] |
| | R | TCTGTTGGTTGGCAGGGACT | | | |
| mx1 | F | TGATCGATAAAGTGACTGCATTCA | SSU66477 | 80 | [64, 68] |
| | R | TGAGACGAACTCCGCTTTTTCA | | | |
| ifih1 | F | GAGAGCCCGTCCAAAGTGAA | XM_014164134 | 389 | [43] |
| | R | TCCTCTGAACTTTCGGCCAC | | | |

ef1a - elongation factor 1 alpha, 18s - 18S ribosomal RNA, ifna1 –interferon a1, isg15—interferon-stimulated gene 15, mx1—interferon-induced GTP-binding protein Mx1, ifih1—interferon induced with helicase C domain 1

S1 Table in S1 File. Transcripts were assembled on the existing Atlantic salmon annotation file (GCF_000233375.1_ICSASG_v2_genomic.gff). Both files were downloaded from NCBI (Annotation release 100). After mapping and assembly of full and partial transcripts, the R package Deseq2 (version 1.22.1) was used to quantify differential expression between groups and against the control [71, 72]. Gene expression tables from Deseq2 were cleaned using median > 10 as a cut off, to get rid of genes with zero or low counts. Using Benjamin-Hochberg (BH) correction to calculate adjusted p-value (padj), genes with p-value (padj) below 0.01 were regarded as differentially expressed genes (DEGs). For gene regulation, was considered upregulated genes with log2 fold change (Log2FC) > 1 and downregulated genes with Log2FC < -1. Exploratory plots and sample analysis is shown in S1-S6 Figs in S1 File. R package clusterProfiler [73] was used for both gene ontology (GO) and KEGG pathway analysis with 0.01 as pvalue and qvalue cutoffs, BH adjusted. Pathview R package was used to draw KEGG pathway maps [74].

## Results

### Effect of EPA on cell viability

To verify that various levels of EPA in the cell culture medium was reflected in cellular membrane lipids, samples from medium and cells were analyzed for EPA as described. Fig 1 shows that cellular EPA content increased proportionally to the levels in the medium, suggesting that the EPA-BSA complex ensured efficient delivery to the cells. We also monitored cell viability using 3 different methods at the same levels of EPA and observed a minor reduction of total viable cells (as measured by neutral red uptake or adherent cells (crystal violet staining)) at the highest levels, but a dose dependent inhibition of resazurin reduction to fluorescent resorufin (Alamar blue assay) by EPA (S7 Fig in S1 File). This may suggest that the highest EPA levels interfere with the cellular levels of mitochondrial or cytoplasmic reductases, the enzymes responsible for Alamar blue reduction to the fluorescent metabolite.

### Effect of poly I:C

To verify the bioactivity of poly I:C, we quantified a set of transcripts previously shown to be induced by poly I:C or interferon [75, 76] by QPCR. Fig 2 shows that these transcripts were strongly induced by the poly I:C treatment and that 200 uM of EPA (highest dose) in itself did not affect the transcription of these genes.

### Effect of different levels of EPA on poly I:C induced transcription

To get a nonbiased analysis on transcriptional effects of EPA on innate immune responses in the Atlantic salmon head kidney cells, we performed RNAseq analysis on samples from cells incubated for one week in various levels of fatty acid, before stimulation with poly I:C. An exploratory analysis of the RNAseq data is presented in S1 File. Briefly, each sample was sequenced to 20 million reads depth and were mapped to a rate of 83%, on average (S1 Table 1 in S1 File). Distribution of counts were lognormal and comparable in all samples with an average of 837 counts per transcript (S1 Fig in S1 File) and A2). Replicate correlation was tested by plotting 100 random transcripts and calculate coefficients. For every sample group the correlation coefficients were above 0.95 (S3 Fig in S1 File). Principal component analysis and hierarchical clustering revealed that poly I:C was the main driver of variability but level of EPA was also discriminatory (S4-S5 Figs in S1 File). Comparing QPCR results with rnaseq also validated the analysis (S6 Fig in S1 File). Plotting of the loadings for the most variable transcripts

**Fig 1. Total levels of EPA (in % of total FA) in medium and ASK cells at various levels of added BSA-EPA (0–200 uM) in the cell medium after one week of incubation.**

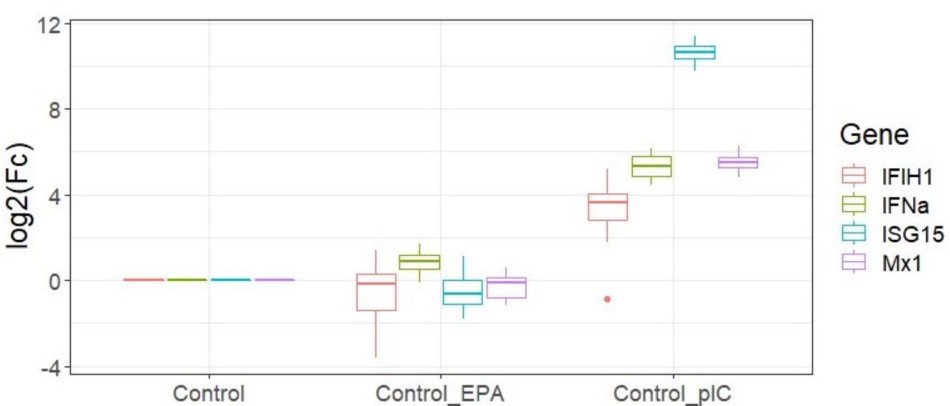

**Fig 2. QPCR analysis of relative expression of interferon inducible transcripts in ASK cells treated with 200 uM EPA for 7 days (Control_EPA) or 24 hours with 30 ug/ml polyI:C (Control_pIC).** Expression was calculated relative to Ef1a and 18S as described. Boxplot shows median, max and min values (n = 6).

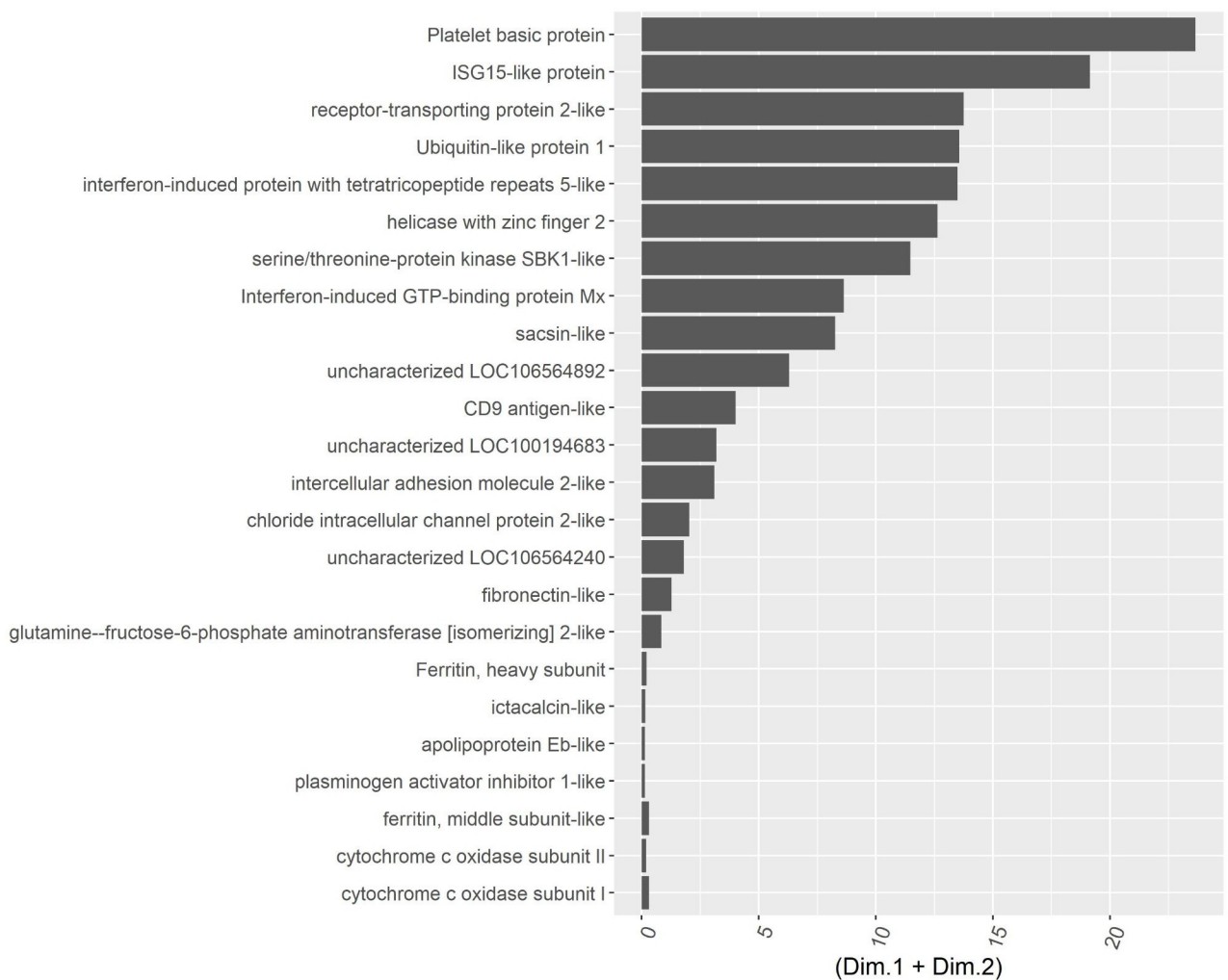

**Fig 3. Principal component loadings for PC1 and PC2 of the most variable transcripts in the RNAseq dataset from Atlantic salmon kidney cells stimulated with poly I:C.**

revealed several well documented interferon regulated genes (ISG15, IFIT5, Mx) supporting that poly I:C had the expected effects (Fig 3).

When cells were incubated with the highest EPA concentration (200 uM) for one week with no subsequent poly I:C stimulation, RNAseq analysis showed only a few perturbed transcripts compared to the effects of poly I:C (104 significant DEGs compared to 4748 for poly I:C). This was not enough to demonstrate systematic effects via enrichment analyses (GO and KEGG). When testing for effects of EPA level on expression of transcripts after poly I:C stimulation we used the recommended likelihood ratio test. According to the Deseq2 vignette: "*The LRT is therefore useful for testing multiple terms at once, for example testing 3 or more levels of a factor at once, or all interactions between two variables. The LRT for count data is conceptually similar to an analysis of variance (ANOVA) calculation in linear regression, except that in the case of the Negative Binomial GLM, we use an analysis of deviance (ANODEV), where the deviance captures the difference in likelihood between a full and a reduced model*" [72]. This test revealed that a total of 3421 transcripts were differentially expressed (DEG) under various EPA levels (p.adjust < 0.01). However, the effects of EPA on expression were subtle for most genes: 100

transcripts were on average (across all EPA levels) induced more than twice (log2FC > 1) whereas more than 1100 transcripts were downregulated more than twofold (log2FC < -1) suggesting that adding EPA to the cell culture medium had predominantly anti-inflammatory effects (attenuating the effect of poly I:C) (S3 Table in S1 File). When averaging the effects of the 4 levels (25–200 uM) of EPA upon poly I:C stimulated transcripts, several immune related transcripts like TNF, Scavenger receptor A, TF Sox-9-A like and others were attenuated. However, transcription of immune related transcripts like lectin and complement factor were also increased, so the effects of EPA on transcripts related to inflammation were bidirectional (Fig 4).

Visualizing the overlap of DEGs in the experimental groups by a Venn diagram revealed an interesting feature of the EPA effects: although EPA alone had only minor effects (104 DEGs), the combination of EPA and poly I:C induced dysregulation of more than 2500 transcripts not perturbed by poly I:C alone (Fig 5).

Pathway enrichment analysis of DEGs affected by the presence of EPA in the medium revealed that many of the affected genes were involved in pathways like PPAR, apelin (S9A Fig in S1 File) autophagy (S10A Fig in S1 File), ferroptosis and fatty acid metabolism (Fig 6). Also enzymes involved in O-glycan biosynthesis were affected. Downregulated transcripts were enriched in the apelin, autophagy and VEGF signaling pathways. In the apelin pathway, transcripts for signaling proteins like protein kinase C, Ras and AMPK were downregulated.

The analysis of deviance described above will identify transcripts that vary across the different EPA concentrations, irrespective of any biological dose response. To identify transcripts displaying consistent positive or negative dose effects of EPA, the coefficients and statistics for expression raw counts correlation to EPA concentration were calculated (using the lm function in R, linear regression). Figs 7 (positive correlation) and 8 (negative correlation) display the poly I:C induced transcripts most sensitive to changes in levels of EPA. The level of DNA-damage-inducible transcripts 4-like protein increased 38 counts for every uM increase in EPA, being the most sensitive transcript with positive correlation. A fatty acid ligase and PPAR gamma-like receptor were also induced by increasing doses of EPA. In contrast, fucolectin-6-like.1 decreased 81 counts for every uM increase in EPA, being the transcript with the strongest negative correlation to EPA levels. Cell adhesion proteins and Krueppel-like factor 2 were also suppressed in a linear fashion.

## Discussion

As marine fats are slowly replaced by oils of vegetable origin in diets, it is of interest to investigate the effects of this change on fish physiology. Some of these effects will be evident during normal conditions, whereas others may not manifest before the animals or cells are subjected to some forms of environmental stress like changes in temperature, infections or crowding [77]. The levels of EPA in the cells after one week on incubation in medium containing up to 200 uM EPA were comparable to tissue levels observed in vivo in salmon fed a diet rich in marine oils [78]. Both QPCR and RNAseq confirmed the strong pro-inflammatory effects of poly I:C on Atlantic salmon cells in vitro by robust upregulation of key interferon regulated genes [43]. Some of the most sensitive transcripts were induced more than a thousand fold within 24 h of exposure to poly I:C. Transcripts like CXC chemokine 11, rsad2, ISG15 and TRIM39 like protein have all previously been confirmed as transcripts induced by virus infection or poly I:C [43]. However, incubation of cells with 200 uM EPA alone had a mild effect on transcriptional programs in these cells. About 100 DEGs were observed, and among these most were changed less than 10 fold. This was also confirmed by GO and KEGG enrichment analysis as no significant enrichments were observed in this group. This is in line with

**Fig 4. The 50 most up- and downregulated transcripts (meanlog2FC across 4 doses of EPA relative to no EPA added) in ASK cells after poly I:C stimulation.**

transcriptomic studies on macrophages from salmon fed different levels of n-3 FA, where only 14 transcripts were affected by diet alone [79]. This lack of effect may be cell type dependent, as previous transcriptomic studies of salmon tissues (liver and muscle) revealed significant correlations between fatty acid levels (of EPA and DHA) and gene expression [78]. Transcripts related to carbohydrate metabolism and insulin signaling were positively correlated to EPA levels in mature salmon liver and muscle.

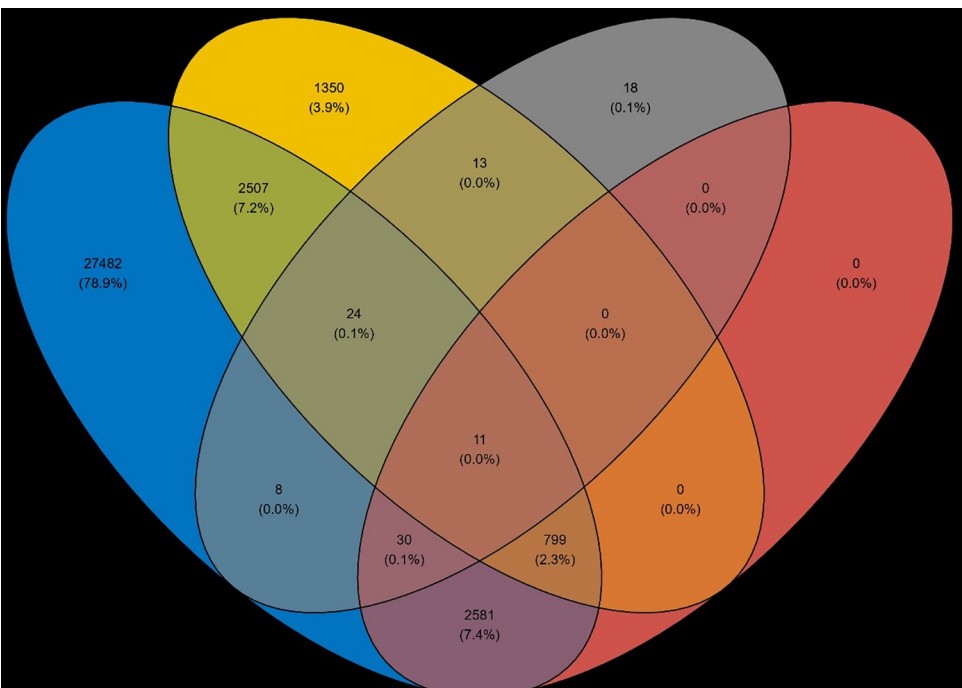

**Fig 5. Venndiagram showing overlap of significant DEGs in the various experimental groups (control are just expressed genes).**

However, in the cell type investigated here, cellular levels of EPA had significant effects on multiple transcriptional programs in the presence of the viral mimic poly I:C. Enrichment analysis of upregulated gene sets identified pathways of general energy metabolism (fatty acid metabolism, fatty acid degradation and PPAR signaling pathway), suggesting that the highest levels of EPA induce altered fatty acid metabolism in these cells. The robust upregulation of perilipin (a target of PPARgamma) may suggest that these cells either store the surplus of lipids or mobilize fatty acids for energy metabolism when stimulated with poly I:C (in the presence of high EPA levels) [80, 81]. PPARs were originally described as master regulators of lipid metabolism, but this family of nuclear receptors possess multiple non-canonical functions like regulation of inflammation [82]. One relevant example is the negative regulation of interferon-beta production in cells and mice upon TLR3 and TLR4 stimulation via repression of IRF3 binding to the IFN beta promotor [83]. In addition, PPAR-gamma deficient macrophages display an increased pro-inflammatory phenotype upon LPS stimulation, suggesting an important role for these receptors as a negative feedback control system during inflammation [84]. This intersection between metabolism and inflammation is now exploited through multiple studies exploring the use of PPAR ligands as antivirals [85].

Apelin is a cytokine secreted from adipose tissues and is the endogenous ligand for G protein coupled APJ receptor [86]. The apelin receptor signaling pathway is active in many cell types and with many tissue dependent effects (contractility, angiogenesis, hemodynamics in addition to enhancing glucose uptake and inhibition of lipolysis [87]. The profound inhibition multiple transcripts this pathway in poly I:C stimulated cells at high EPA levels may suggest that apelin targets are reduced during inflammation. Activation of the apelin pathway protects lipid droplets from fragmentation and lipolysis [88]. One possible explanation for the observed downregulation could therefore be to attenuate this protection, enhancing the PPAR gamma dependent mobilization of fatty acids during the inflammatory response [89].

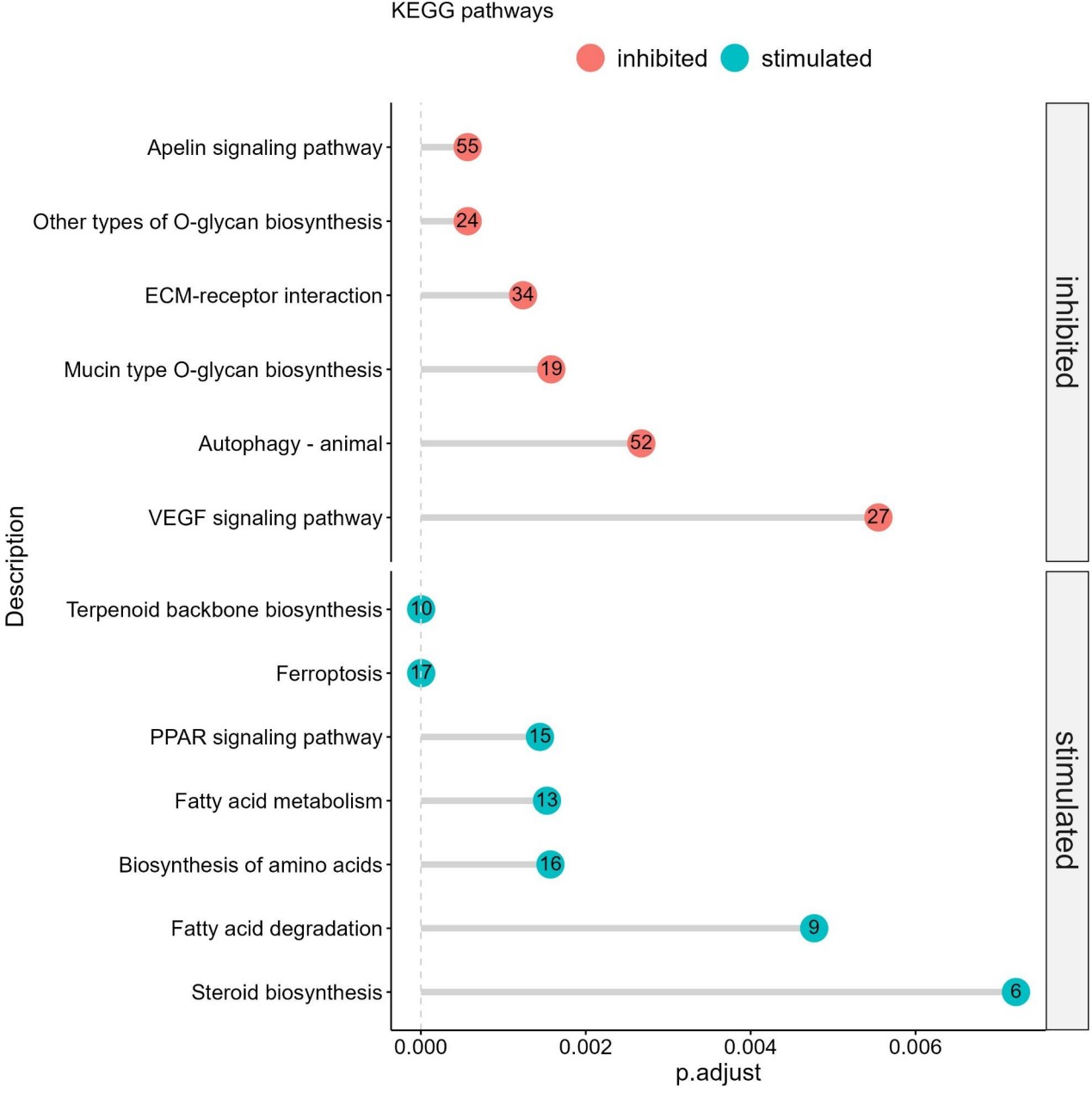

**Fig 6. Enriched KEGG pathways in ASK cells incubated in EPA and stimulated with poly I:C.**

The enrichment of downregulated transcripts related to autophagy may suggest that EPA reduce inflammatory processes in poly I:C stimulated cells. Autophagy plays a key role in protein and energy metabolism during starvation but is also involved in stress responses due to hypoxia, ER-stress, reactive oxygen radicals as well as viral infections [90]. It is well documented that autophagy play a role both as antiviral defense mechanism [91] as well as in controlling the response by selective degradation of interferon regulatory factor 3 [92]. The enrichment of upregulated transcripts in the ferroptosis pathway may indicate that these high levels of cellular EPA in combination with a strong interferon inducer like poly I:C, induce generation of lipid peroxide formation [93]. The combination of these factors with the

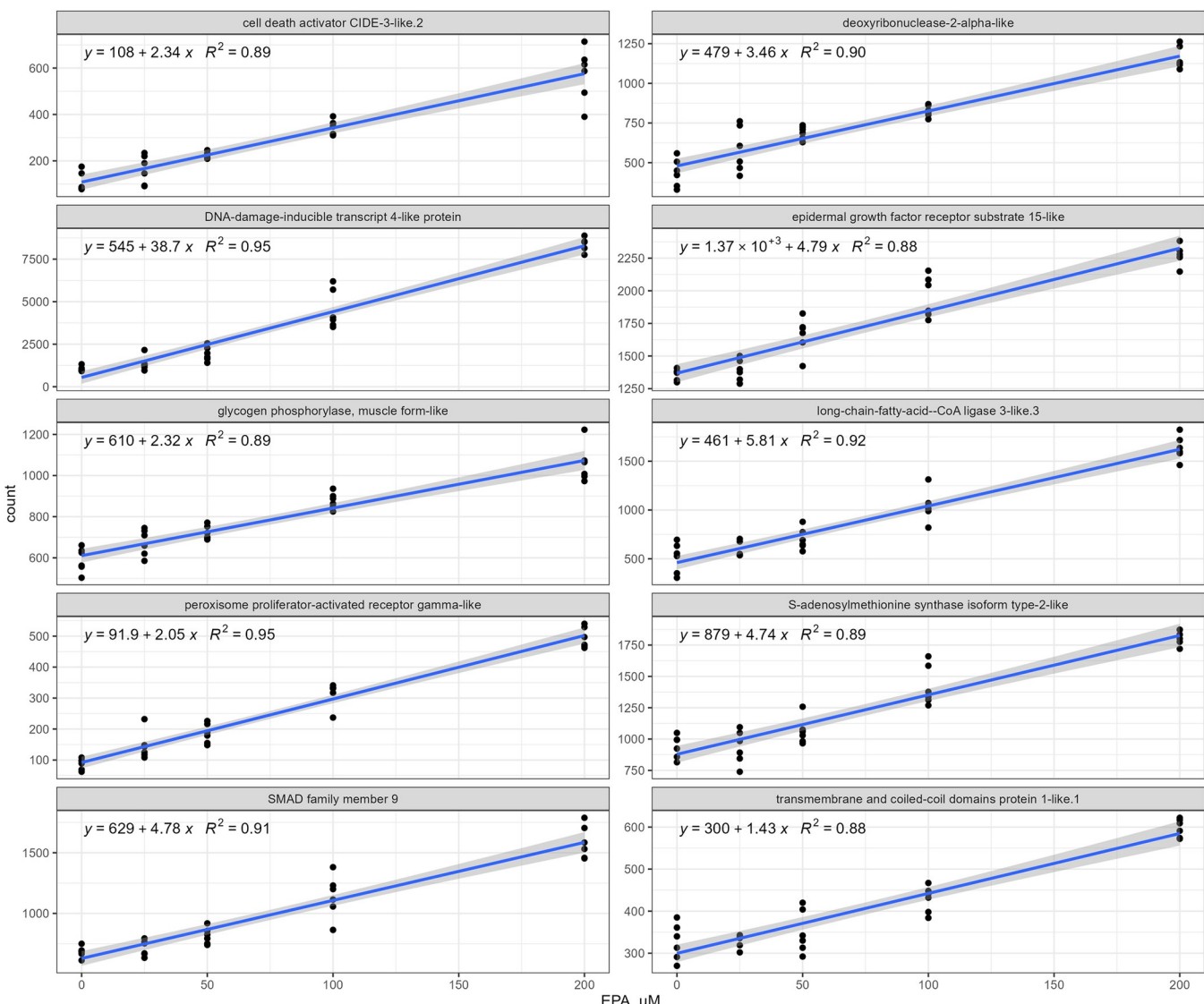

**Fig 7. Plot of raw counts and best fit line for the transcripts most positively correlated to EPA concentration in poly I:C stimulated ASK cells.**

observation that ASK cells express high levels of ferritin mRNA (suggestive of high iron load) may lead to ferroptosis [94]. Ferroptosis can be described as a regulated nonapoptotic cell death caused by iron-dependent accumulation of reactive oxygen species (ROS) [95]. Ferroptosis can be induced by blockage of the cysteine/glutamate antiporter system $X^-_c$ (SLC3A2/SLC7A11) leading to reduced cellular glutathione levels (and therefore reduced antioxidant capacity) [96]. The downregulation of this complex in the EPA/poly I:C treated cells combined with upregulation of the transferrin receptor (hence increased iron uptake) may therefore explain the enrichment of ferroptosis related transcripts observed here. Interestingly, synergistic effects between low dietary iron and high omega-3 fatty acids related to iron storage capacity were observed during natural outbreaks of bacterial disease in Atlantic salmon [97].

In vitro studies like this using immune stimulants like poly I:C does not tell the full story about fish innate immunity to viral infections under various dietary conditions. However, the experimental design ensure a synchronous stimuli to all cells, and therefore increase the

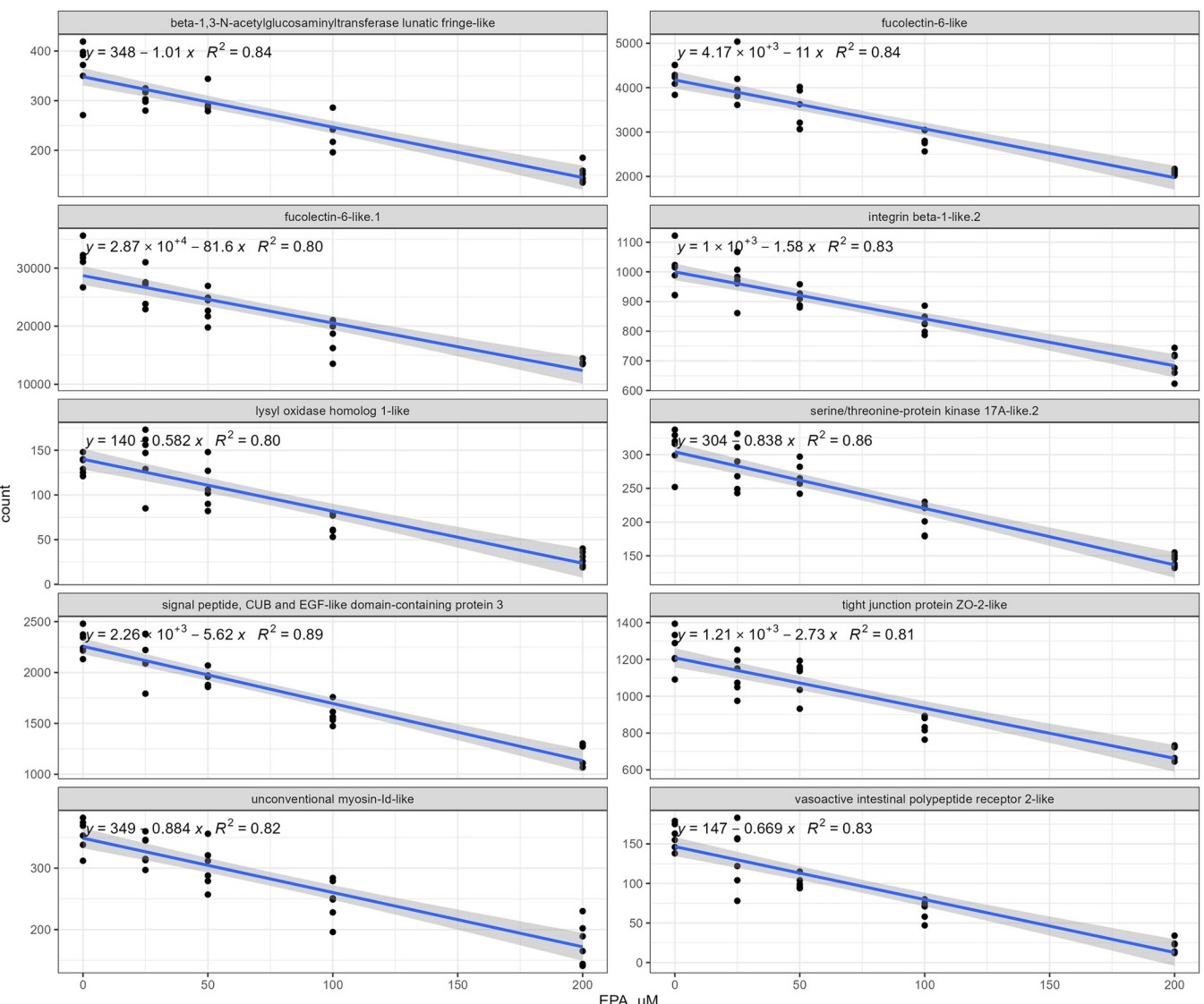

**Fig 8. Plot of raw counts and best fit line for the transcripts most negatively correlated to EPA concentration in poly I:C stimulated ASK cells.**

chance to detect weak transcriptional signals normally induced asynchronously during viral infections in vitro or in vivo [98]. In addition, we [43] and others have observed substantial overlaps in the transcriptional responses between poly I:C and viral infections in cells from salmon and other fish species, ensuring its relevance as a model. In addition, the development of new and safe fish vaccines with less side effects (like oil induced visceral adhesions [99]) includes poly I:C as one possible new adjuvant candidate for fish vaccines [100]. In parallel, as new feeds based on plant material are replacing EPA rich fish oils in the production of fish feeds, the impact of these changes on immunity need more scrutiny [101]. In one study we found that varying levels of EPA/DHA do impact the early innate responses to vaccination in Atlantic salmon and that adaptive responses were more resilient to perturbations [102]. Mechanistic studies of cellular responses to new vaccines and feeds in aquaculture is therefore relevant for improved fish health and welfare [103, 104]. However, extending these studies with additional in vivo feeding studies combined with vaccination will be necessary to evaluate the

impact of feeding ingredient on immunity. One conclusion from this work was that increasing levels of cellular EPA had a significant impact on the transcriptional programs activated by the viral mimic, poly I:C. This is in line with previous reports on the intimate interplay between general metabolism and inflammation. What this means for antiviral immunity in live fish remains to be elucidated in further studies using challenge models of viral infection.

## Supporting information

**S1 File. This file contains supplementary figures and tables.**
(PDF)

## Acknowledgments

We thank Beata Mohebi, Truls Rasmussen and Målfrid Bjerke for skillful technical assistance.

Data reporting–raw data for the rnaseq analysis have been deposited at the SRA archive (https://www.ncbi.nlm.nih.gov/sra) with accession # PRJNA1033415.

## Author Contributions

**Conceptualization:** Tor Gjøen.

**Data curation:** Tor Gjøen.

**Formal analysis:** Tor Gjøen.

**Funding acquisition:** Tor Gjøen, Bente Ruyter, Tone Kari Østbye.

**Investigation:** Tor Gjøen, Bente Ruyter, Tone Kari Østbye.

**Methodology:** Tor Gjøen.

**Supervision:** Tor Gjøen.

**Validation:** Tor Gjøen.

**Writing – original draft:** Tor Gjøen.

**Writing – review & editing:** Bente Ruyter, Tone Kari Østbye.

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
