## [Decision Letter · Decision Letter 0]

12 Jan 2024

PONE-D-23-38632Effects of eicosapentaneoic acid on innate immune responses in an Atlantic salmon kidney cell line in vitroPLOS ONE

Dear Dr. Gjøen,

Thank you for submitting your manuscript to PLOS ONE. After careful consideration, we feel that it has merit but does not fully meet PLOS ONE’s publication criteria as it currently stands. Therefore, we invite you to submit a revised version of the manuscript that addresses the points raised during the review process.

We look forward to receiving your revised manuscript.

Kind regards,

Amit Ranjan, Ph.D.

Academic Editor

PLOS ONE

Journal Requirements:

https://link.springer.com/article/10.1007/s10695-017-0353-4?

In your revision ensure you cite all your sources (including your own works), and quote or rephrase any duplicated text outside the methods section. Further consideration is dependent on these concerns being addressed."""

AAM (Straive) 28 Nov 2023: IRB for Animal Not Stated

To comply with PLOS ONE submissions requirements, please provide the following information in the Methods section of the manuscript and in the “Ethics Statement” field of the submission form (via “Edit Submission”):

* Please indicate whether an animal research ethics committee prospectively approved this research or granted a formal waiver of ethics approval.* Please enter the name of your Institutional Animal Care and Use Committee (IACUC) or other relevant ethics board. Also include an approval number if one was obtained.

* If anesthesia, euthanasia, or any kind of animal sacrifice is part of the study, please include briefly in your statement which substances and/or methods were applied.

For additional information about PLOS ONE submissions requirements for ethics oversight of animal work, please refer to http://journals.plos.org/plosone/s/submission-guidelines#loc-animal-research

" this study was supported via grant #901484 from the Norwegian Seafood Research fund (FHF). We thank Beata Mohebi, Truls Rasmussen and Målfrid Bjerke for skillful technical assistance."

"TG, BR and TKØ received grant #901484 from the Norwegian Seafood Research fund (FHF)

https://www.fhf.no/

The sponsor played no role in the experimental design"

6. Please ensure that you refer to Figure 8 in your text as, if accepted, production will need this reference to link the reader to the figure.

Reviewers' comments:

Reviewer's Responses to Questions

**Comments to the Author**

1. Is the manuscript technically sound, and do the data support the conclusions?

Reviewer #1: Yes

Reviewer #2: Yes

2. Has the statistical analysis been performed appropriately and rigorously? 

Reviewer #1: Yes

Reviewer #2: Yes

3. Have the authors made all data underlying the findings in their manuscript fully available?

Reviewer #1: Yes

Reviewer #2: Yes

4. Is the manuscript presented in an intelligible fashion and written in standard English?

Reviewer #1: Yes

Reviewer #2: Yes

5. Review Comments to the Author

Reviewer #1: Summary

The article discusses the field of immunometabolism, focusing on the interaction between nutrients, energy metabolism, and the immune system in both health and disease. It emphasizes the limited understanding of the interplay between feed and immunity in fish, particularly in the context of dietary changes in Atlantic salmon. The traditional dietary sources of essential fatty acids, such as fish oil and fish meal, are being substituted by plant oils, impacting the levels of eicosapentaenoic acid (EPA) and docosahexaenoic acid (DHA) in salmon diets. The study explores the effects of different cellular levels of EPA on the transcriptional responses of salmon cells to the viral mimic poly I:C, revealing a potential link between EPA levels and the modulation of immune responses. The article is well-written, scientifically rigorous, and contributes valuable insights to the field of fish immunometabolism.

General Comments

I acknowledge the limitations of generalizing findings from Atlantic salmon cells to other fish species or different populations of Atlantic salmon. Explicitly discuss how the observed changes might impact the overall well-being and performance of fish.

• The article primarily focuses on EPA, but it would be valuable to explore potential interactions with docosahexaenoic acid (DHA) since fish oil traditionally contains both EPA and DHA (Lines 68-69).

• The reliance on in vitro cell culture experiments is noted. Consider addressing the limitations of the cell culture model and discussing how in vivo studies could complement these findings (Lines 110-112, 368-370).

• The study assesses short-term effects after one week. Consider discussing potential long-term effects or the sustainability of observed changes over extended periods (Lines 114-124).

Recommendation

I recommend this manuscript for publication in Plos One. The study opens avenues for further research on the role of EPA in antiviral immunity in live fish and highlights the scarcity of similar studies in fish.

Reviewer #2: Dear authors,

Congratulations and you have done a good job. I have critically gone through the manuscript entitled “Effects of eicosapentaneoic acid on innate immune responses in an Atlantic salmon kidney cell line in vitro”. However, you need to incorporate the revision and answer the comments that are stated below before it is published.

1. Abbreviation should be expanded while writhing for the first time.

2. L1 write the scientific name of Atlantic salmon.

3. L28 and L30 write the full form of VEGF and PPAR.

4. L42 write the full form of RLR signalling.

5. Broken line analysis may use to know the exact EPA level rather than writing it as 200 uM EPA. It may be less or more than 200 uM EPA

6. PLOS authors have the option to publish the peer review history of their article (what does this mean?). If published, this will include your full peer review and any attached files.

Reviewer #1: No

Reviewer #2: **Yes: **Gyandeep Gupta

---

## [Author Response · Author response to Decision Letter 0]

14 Mar 2024

Thank you for a thorough review of our ms. A separate letter with comments to your questions is included

---

## [Decision Letter · Decision Letter 1]

2 Apr 2024

Effects of eicosapentaneoic acid on innate immune responses in an Atlantic salmon kidney cell line in vitro

PONE-D-23-38632R1

Dear Dr. Gjøen,

We’re pleased to inform you that your manuscript has been judged scientifically suitable for publication and will be formally accepted for publication once it meets all outstanding technical requirements.

Kind regards,

Amit Ranjan, Ph.D.

Academic Editor

PLOS ONE

Reviewers' comments:

Reviewer's Responses to Questions

**Comments to the Author**

1. If the authors have adequately addressed your comments raised in a previous round of review and you feel that this manuscript is now acceptable for publication, you may indicate that here to bypass the “Comments to the Author” section, enter your conflict of interest statement in the “Confidential to Editor” section, and submit your "Accept" recommendation.

Reviewer #1: All comments have been addressed

Reviewer #3: All comments have been addressed

2. Is the manuscript technically sound, and do the data support the conclusions?

Reviewer #1: Yes

Reviewer #3: Yes

3. Has the statistical analysis been performed appropriately and rigorously? 

Reviewer #1: Yes

Reviewer #3: Yes

4. Have the authors made all data underlying the findings in their manuscript fully available?

Reviewer #1: Yes

Reviewer #3: Yes

5. Is the manuscript presented in an intelligible fashion and written in standard English?

Reviewer #1: Yes

Reviewer #3: Yes

6. Review Comments to the Author

Reviewer #1: The authors have addressed all the comments; therefore, the manuscript can be accepted for publication.

Reviewer #3: This study investigated various cellular EPA levels modulate the antiviral transcriptional responses in salmon cells during stimulation with poly I:C, and the results suggested that innate antiviral responses are heavily influenced by the fatty acid profile of salmonid cells and constitute another example of the strong linkage between general metabolic pathways and inflammatory responses.

The manuscript had been well prepared and could be considered for publication.
